



# Introducing a data-driven approach to predict site-specific leading edge erosion

Jens Visbech[1], Tuhfe Göçmen[1], Charlotte Bay Hasager[1], Hristo Shkalov[2], Morten Handberg[2], and Kristian Pagh Nielsen[3]

[1]Department of Wind and Energy Systems, Technical University of Denmark (DTU), 4000 Roskilde, Denmark
[2]Wind Power LAB, 1150 Copenhagen, Denmark
[3]Danish Meteorological Institute (DMI), 2100 Copenhagen, Denmark

**Correspondence:** Jens Visbech (jvima@dtu.dk)

**Abstract.** Modeling leading edge erosion has been a challenging task due to its multidisciplinary nature involving several variables such as weather conditions, blade coating properties, and operational characteristics. While the process of wind turbine blade erosion is often described by engineering models that rely on the well-known Springer model, there is a glaring need for modeling approaches supported by field data. This paper presents a data-driven framework for modeling erosion damage

based on blade inspections from several wind farms in Northern Europe and mesoscale numerical weather prediction (NWP) models. The outcome of the framework is a machine-learning based model that can be used to predict and/or forecast leading edge erosion damages based on weather data/simulations and user-specified wind turbine characteristics. The model is based on feed-forward artificial neural networks utilizing ensemble learning for robust training and validation. The model output fits directly into the damage terminology used by industry and can therefore support site-specific planning and scheduling of

repairs as well as budgeting of operation and maintenance costs.

## 1 Introduction

Wind energy is still deemed one of the most important solutions to combat climate change and it is foreseen that the annual installation of new wind turbines will reach 280 GW in 2030 (Council, 2021). Offshore wind energy, in particular, is being favored mainly because of the higher wind resource and the limited socio-environmental impact. In addition, even larger wind

turbines are being introduced on the market with bigger rotors and higher hub heights. To ensure structural compliance of such huge, moving structures, the increasing demand for turbines with higher power capacity will inevitably require the turbines to operate with higher rotational speed (Ning and Dykes, 2014; Dykes et al., 2014). Rotor speed is directly proportional to the blade tip speed and modern turbines are typically operating with a maximum tip speed in the range of 80-100 m/s. For onshore wind turbines, the maximum tip speed is often lower than that of offshore wind turbines because of strict regulations

on acoustic noise emission. With wind turbine blades moving at such a high speed, the impact from particles like rain, hail, or sand, has a devastating effect on the leading edge of the blade. Repetitive impact eventually causes the surface coating to erode. This phenomenon of leading edge erosion has been and still is a significant financial uncertainty in the economic planning of wind turbines.



The development and progression of leading edge erosion are constituted of different stages. Initially, an incubation period
occurs where no or little damage is visible on the surface. After the incubation period, the erosion rate accelerates until it
reaches a maximum erosion rate. Eventually, the erosion rate decelerates into a steady erosion state. The duration of each stage
is mainly affected by the strength properties of the coating material and the weather conditions it is exposed to (Springer et al.,
1974).

The effects of leading edge erosion have two main contributions to the financial viability of the wind turbine. Firstly, it
affects the aerodynamic properties of the blade. Small changes in the blade surface geometry affect the lift and drag forces
thereby reducing the energy production. Quantifying and validating the annual energy production (AEP) loss from erosion is
highly challenging as it depends on the extent and severity of erosion along the leading edge. Several studies have investigated
the AEP loss through wind tunnel testing and numerical simulations, and results have shown highly varying losses in the range
of 1-25 % (Cappugi et al., 2021; Ehrmann et al., 2017; Sareen et al., 2014; Kruse et al., 2021b, a; Herring et al., 2019).

Secondly, without preventive actions or regular maintenance, the erosion damage eventually reaches a state of severity
where the structural integrity of the blade is threatened. At this stage, the only option is to repair or replace the wind turbine
blade. Comprehensive repair campaigns have been conducted on offshore wind farms, e.g., Anholt wind farm in Denmark
and London Array in the United Kingdom. Such repair campaigns are extremely costly, especially offshore, because of high
service prices and lost production during downtime. It has been reported that unplanned repairs from minor failures, such as
erosion, occur 12 times more often than that from structural failure (Mishnaevsky Jr and Thomsen, 2020). Existing repair
solutions are classified according to the erosion severity, affected region, and aerodynamic requirements (Mishnaevsky Jr,
2019). They include gel coats, flexible coatings, leading edge tapes, and external or integrated erosion shields, and they all
have advantages and disadvantages related to erosion resistance, cost, application, and debonding risk. In addition, solutions
might also negatively influence the aerodynamic properties and are often subject to a requirement of continuous maintenance
(Herring et al., 2019).

While a lot of research has previously been on the prediction of the negative impact that leading edge erosion has on
AEP, there is an increasing need for models that can be used for the prediction of erosion damages. These erosion damage
prediction models can be used for making adequate repair planning, thereby assisting the overall assessment and management
of operational expenditure (OPEX). There exist a number of different studies, proposing models for estimating the fatigue
lifetime of a blade being exposed to erosive conditions. Bech et al. (2018) proposed an erosion-safe mode (ESM) operational
strategy to mitigate leading edge erosion by reducing the rotational speed during heavy rainfalls. The control strategy relies
on empirically derived SN-curves of different coatings obtained from rain erosion tests. These properties are used to estimate
the fatigue life of a coating for given time series of the wind and rain climate. The model is then used for cost optimization to
determine the best control strategy and repair schedule.

Prieto and Karlsson (2021) proposed a model to characterize the severity of erosion using a multidisciplinary model by
combining meteorological data, operational settings, and structural properties. The model takes its basis in the well-known
analytical surface model from Springer (1976) and considers four different types of erosion mechanisms, namely rainfall,



snow, sea spray, and fog. The model is validated against five reference wind farms and shows a good qualitative comparison of model-predicted erosion incubation and experimental observations.

Probabilistic approaches such as that from Verma et al. (2021a) and Verma et al. (2021b) have also been proposed for erosion prediction. These studies rely on joint probability distributions of rain features (rate, droplet size) and wind speed as inputs to an analytical surface fatigue model that outputs the expected lifetime of a blade coating. The probabilistic framework was used to evaluate the erosion climate at different sites in the Netherlands and showed that especially coastal sites were prone to more severe weather conditions and required more frequent repairs. However, the proposed erosion model lacks validation via actual

blade inspections on the field.

   While the above-mentioned engineering models rely heavily on theoretical approaches, there also exists a proposition of data-driven methods utilizing machine learning for predictive modeling. An advantage of this type of model is the ability to map complex relations entirely based on the data, i.e., without a need for prior physical knowledge. In addition, predictive modeling allows for easy adaptation of new incoming data.

Martinez et al. (2019) trained a boosted random forest regressor using inspection data from 17 different wind farms in North America. The model target was blade defects, grouped into five severity classes, and predicted using time-aggregated weather and operational features such as rain, wind speed, and operational information. The developed methodology, however, has a strong dependency on the availability of rain measurements and SCADA signals from the site. Accordingly, it has limited use for the majority of the offshore locations (where rain/precipitation is typically not measured) as well as time-ahead damage

predictions and repair scheduling (including the design stage of wind farms).

   A recent study from Castorrini et al. (2021) introduces a more complex implementation of machine learning using a lookup table approach for predicting rain erosion damage. The erosion model takes operational settings and environmental conditions for a blade section and estimates the distribution of erosion damage on the blade section. Although the methodology was qualitatively validated on a reference turbine after one year of operation, there is a need for large-scale validation against

different erosion cases.

   Another study from Cappugi et al. (2021) presents an AEP loss prediction system that utilized artificial neural networks for estimating lift and drag coefficients for eroded wind turbine blades. The neural network is trained using CFD-generated airfoil data for a few erosion classes, enabling fast estimation of the aerodynamic properties. Using blade element momentum theory, the AEP of the NREL 5 MW turbine is estimated for different wind conditions, and the study reported AEP losses up to 4 %.

Duthé et al. (2021) proposed a spatio-temporal stochastic model for generating simulations of leading edge erosion along a wind turbine blade. An attention-based Transformer model was trained on the simulated eroded blades to be able to detect and classify the degradation severity based on multivariate time series. The simulated blades can also be used in a forward aeroelastic simulation software to determine the effects of an eroded blade on the dynamic wind turbine response. As a result of the model being trained entirely on simulated erosion data, there is a lack of field data validating the predicted erosion cases.

Common for many of the existing leading edge erosion models, is their dependence on detailed information about precipitation characteristics, operational data, and coating properties, namely VN-curves obtained from rain erosion tests. Such tests are useful for characterizing the fatigue strength of a coating when exposed to rain in a controlled environment with well-known



conditions. However, translating these engineering models from a controlled environment to the field would require equally accurate information about local precipitation characteristics and operational data. This type of data is rarely available which
urges the need for a model that relies on more realistic data availability.

In this paper, we present a novel machine learning (ML) approach for estimating and forecasting blade defects caused by leading edge erosion. An ensemble model based on simple neural networks is trained using mesoscale weather data and industrial blade inspections from several wind farms across North Europe. Mesoscale weather simulations are preprocessed and used to make them compatible with the overall framework and the blade inspections are encoded using a unique weighting
scheme. Utilizing the liquid rain impingement as the global erosion damage predictor, the model essentially behaves as a complex transfer function that converts a sequence of common weather data into an interpretable estimate of the expected erosion damage. The erosion model can be used to make site-specific erosion predictions based on historical weather data or erosion forecasts based on climatological characteristics. The proposed erosion model does not rely on coating properties or operational data but rather incorporates the inherent variability of leading edge erosion. More specifically, this is done by
considering multiple blades for the encoding, thereby limiting the relative effect of individual blades. While existing models listed earlier lack a reference for their erosion predictions that is consistent and relatable for the industry, the present erosion model fits directly into the terminology used by the industry for making repair recommendations and OPEX budgeting.

The paper is structured as follows: Section 2 describes the overall methodology covering the analysis and preprocessing of meteorological data and blade inspections. This section also describes the entire modeling process from feature selection to
model validation. In section 3, the results obtained throughout the study are presented including model validation, performance, and application. Section 4 contains a discussion of the uncertainties related to the data and model as well as a general discussion of the erosion prediction model. Finally, the main conclusions are summarized in section 5.

## 2   Methodology

The overall workflow of the modeling framework can be found in Figure 1 showing the main steps. The workflow is two-folded
in the way that it starts with the model training process which is governed by the available blade inspections and weather data. The training is followed by the application of the trained model which takes local weather data and user-specified wind turbine characteristics to generate the needed time-aggregated model inputs. The output of this flow is site-specific damage prediction, e.g., in the form of an interactive erosion map. A detailed description of each step in workflow will be given forthwith.

### 2.1   Mesoscale Weather Simulations

The weather data included in both the training and application of the ML-based blade defect forecasting algorithm is from the HARMONIE non-hydrostatic, convection-permitting mesoscale numerical weather prediction (NWP) model (Bengtsson et al., 2017). The HARMONIE model domain provided by DMI has changed over time and in order to cover longer periods, both the smaller DKA domain (Yang et al., 2012) (May 2013 – November 2016) and larger NEA domain (Yang et al., 2017) (November 2016 – January 2021), as shown in Figure 2, are utilized in the analysis. Although the horizontal model grids are

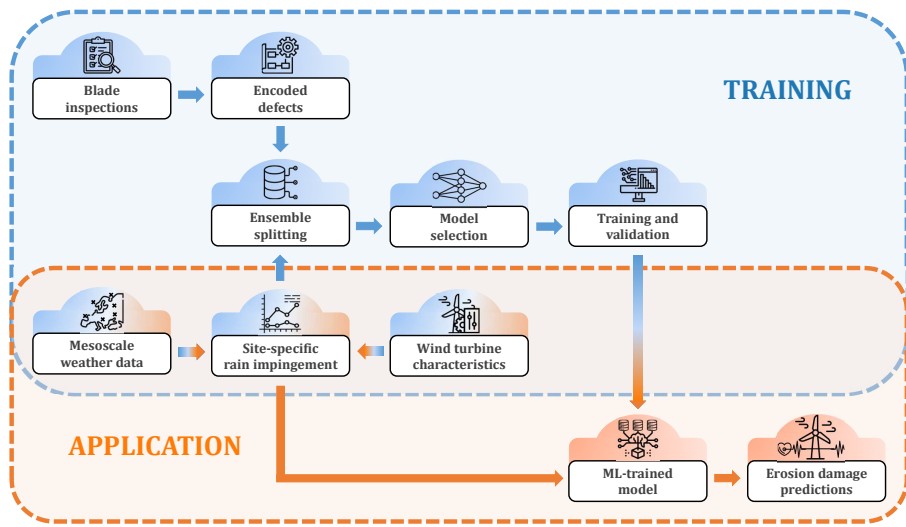

**Figure 1.** High-level workflow visualizing the main steps of the modeling framework and the distinction of training and application flows.

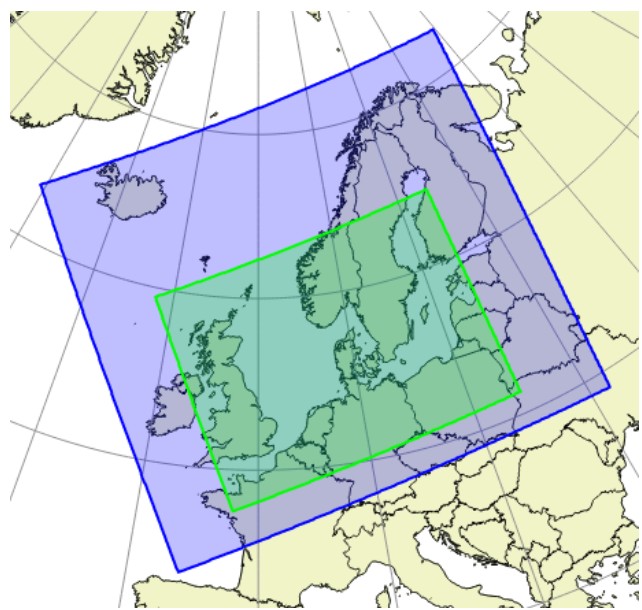

**Figure 2.** An overview of 2 DMI model domains from which data are included. The NEA domain is shown in blue color, and the DKA domain is shown in green color.

different between the NEA and DKA domains, they both have a horizontal resolution of 2.5 km for the provided hourly time series.

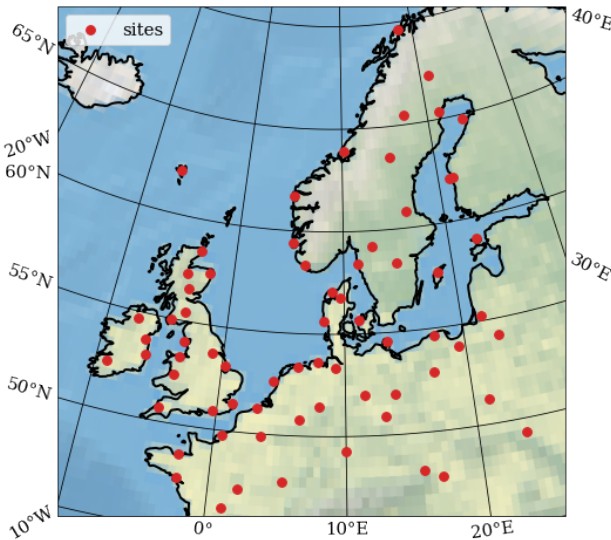

**Figure 3.** Map of Northern Europe indicating the locations of the sites used for training (7 in total, anonymized) and testing (99 in total). Made with Natural Earth.

In total, seven sites are analyzed for the training of the model where coinciding blade inspections are available. After the initial evaluation, the trained model is then used for blade defect predictions among 99 sites across Northern Europe. Their locations are indicated in Figure 3. From the HARMONIE runs, accumulated hourly surface fields are used to identify the precipitation (rain and solid precipitation). For the hourly wind speed components, the model level data extracted closest to the hub height of the turbines for the training sites are utilized. In addition, they are vertically extrapolated to the exact hub heights using the power law with a shear exponent of $\alpha = 0.16$. For the remaining sites to be used for the forecasting of the blade defects (model application stage), the model level height is kept slightly higher than 100 m, given typically larger turbines and higher hub heights in the new and upcoming wind farm sites. For both the training and inference of the data-driven model, hourly time series of the rain and wind speed extracted at the center of the 2.5 km resolution grid, corresponding to the locations indicated in Figure 3, are fed into the impingement model to determine the final input features of the ML algorithm.

The impingement model used in the present study follows a simple approach adopted from (Bech et al., 2018). Essentially, the rain impingement represents the amount of rain (water column) that impinges the tip of the blade. It can therefore be estimated from basic operational settings of the wind turbine and time series of wind speed $U$ and rain intensity $Ir$ following a few simple assumptions.

The relative volume of water in the air can be calculated by:

$$W = \frac{Ir}{V_t} \qquad (1)$$





where $V_t$ is the terminal velocity of the rain droplet. It is approximated using a deterministic approach based on the rain intensity $Ir$ by 1) following the empirical relation between rain intensity and droplet size suggested by (Best, 1950) and 2)

following the empirical relation between droplet size and terminal velocity suggested by (Foote and Du Toit, 1969).

Assuming the rain droplets follow the wind speed, the maximum impact velocity of the droplets perpendicular to the blade at the tip can be calculated by:

$$V_{max} = \sqrt{U^2 + V_{tip^2}} = \sqrt{U^2 + (\omega \cdot R)^2} \tag{2}$$

where $R$ is the blade length and $\omega$ is the rotor speed. The rotor speed is estimated using a very simplistic approach where it is

described using a linear threshold function, i.e., $\omega = 0$ for wind speeds below the cut-in wind speed, $\omega$ increases linearly from minimum to maximum rotor speed between cut-in and rated wind speed, $\omega$ is constant at the maximum rotor speed between rated and cut-out wind speed, and finally, $\omega = 0$ for wind speeds above cut-out wind speed.

The rain impingement is then calculated by:

$$R_{imp} = W \cdot V_{max} \cdot \Delta t = \frac{Ir_i \cdot \sqrt{U_i^2 + (\omega \cdot R)^2}}{V_t} \cdot \Delta t \tag{3}$$

where $\Delta t$ is the length of a time series bin.

Figure 4 shows an example of a 5-year time series of wind speed (top), rain rate (middle), and the corresponding accumulated rain impingement (bottom). If this time series was to be used as a sample, the point value to be used as input would be approximately 40 m. Considering the appearance of the accumulated impingement, it is noticed that the curve appears to be increasing almost linearly. This is a general trend observed for all the considered sites, which indicates that the accumulated

impingement is constituted of many rainy hours with low intensity rather than a few hours of high-intensity rain. For this particular site during the five-year period, heavy rain with intensity higher than 10 mm/hr occurred only 0.02 % of the time, moderate rain with intensity between 2.5 and 10 mm/hr occurred 0.54 % of the time whereas light rain with intensity below 2.5 mm/hr occurred 4.94 % of the time.

To account for any missing periods, the accumulated rain impingement was scaled with a factor corresponding to the ratio

between the theoretical length of the time series and the actual available length of the time series. Generally, the scaling factor was found to be close to one, indicating a good availability. However, one of the wind farms was commissioned in 2009, therefore, the temporal coverage by the mesoscale weather data is only partial and relies on linear extrapolation.

## 2.2 Blade inspections

The blade inspection data used for the study are provided by Wind Power LAB and include inspections of 678 wind turbine

blades from 7 different wind farms located both onshore and offshore in Northern Europe. The size of the inspected wind farms varies greatly with the smallest wind farm containing just two wind turbines and the largest wind farm containing 60 wind turbines. The inspected wind turbines are all manufactured by Vestas, except for a few turbines from Siemens, and they all have nominal capacities between 2-3.6MW. The specific turbine types used in the study are listed in Table 1 below, together with their main operational characteristics considered for this study.



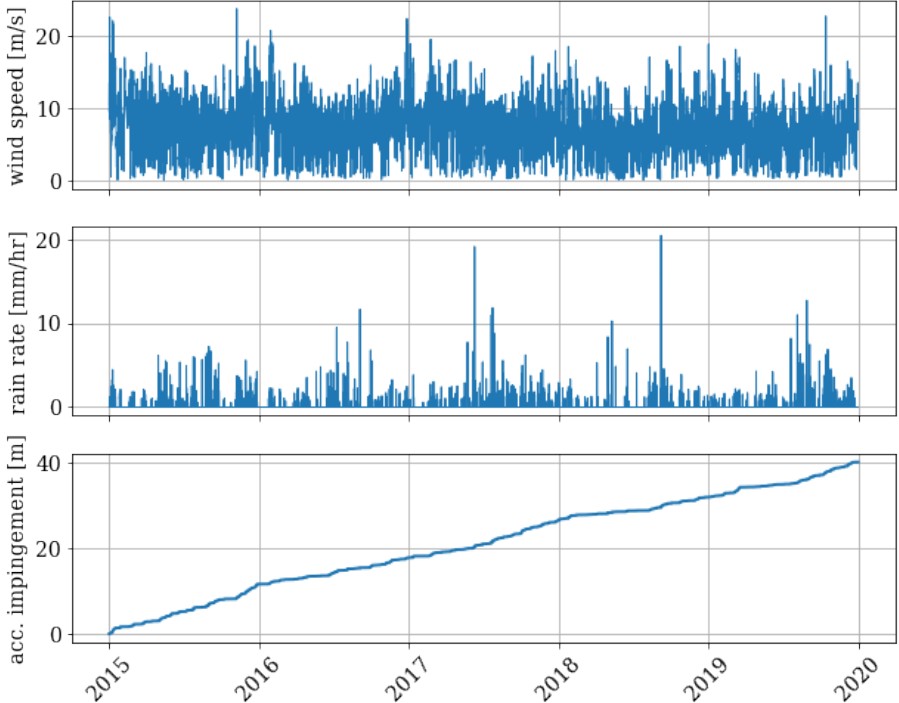

**Figure 4.** Example of a 5-year time series of wind speed (top), rain (middle), and accumulated rain impingement (bottom) from one of the training sites.

**Table 1.** Summary of turbine types and the assumed basic operational characteristics.

|  |  | Vestas V80-2000 | Vestas V90-2000 | Vestas V90-3000 | Vestas V100-2000 | Vestas V126-3450 | Siemens SWT3.6-120 |
|---|---|---|---|---|---|---|---|
| Nominal capacity | [kW] | 2000 | 2000 | 3000 | 2000 | 3450 | 3600 |
| Blade length | [m] | 40 | 45 | 45 | 50 | 63 | 60 |
| Cut-in wind speed | [m/s] | 3.5 | 3 | 3 | 3.5 | 4.5 | 3.5 |
| Rated wind speed | [m/s] | 14.5 | 13.5 | 13.5 | 12 | 11.5 | 14 |
| Cut-out wind speed | [m/s] | 25 | 25 | 25 | 22 | 22 | 25 |
| Minimum rotor speed | [rpm] | 9 | 8.2 | 8.2 | 7 | 5 | 5 |
| Maximum rotor speed | [rpm] | 19 | 17.3 | 17.3 | 13.4 | 13 | 13 |

The blade inspections are provided as tabular data containing a row for each observed defect. For each defect, the vertical position and affected surface area are detected and the defect is categorized by a defect type and a defect severity. The purpose of the defect categorization is to ensure integrity and provide an assessment of the blade condition which allows for making specific repair recommendations. For that reason, each unique defect type and severity is assigned a damage weight that





represents the urgency to repair. The damage weights also ensure the natural defect progression that has been observed from
inspections, i.e., a defect can only progress to a higher weighted defect. The different types of defects and severities are shown
in Figure 5 with the corresponding weights.

Generally, the defect type refers to the specific type of surface failure detected whereas the defect severity generally refers to
the surface layer that is affected but also the effect on structural integrity. Voids are surface coat defects that are characterized
by a pinhole-like appearance and are typically caused by imperfections during the application of the filler and paint. Chipping
often starts at locations where voids have weakened the surface coating allowing chipping to develop. It is characterized by
rough and uneven edges around the defect border. Peeling describes instances where the paint or filler is detached from the
surface material due to poor adhesion, which is usually a result of a poor coating application process. Erosion is a particularly
progressive defect type. It is observed as a continuous strip of roughened surface and is typically characterized by high length-
to-width ratio and continuity. Like chipping, erosion is mainly caused by high-speed impacts on the blade, and therefore, it is
often observed closer to the tip of the blade.

Defect sizes are not a part of the current study, so minor voids exposing surface laminate would be treated in the same way in
terms of repair as a large area of peeling exposing surface laminate if the classical severity model is followed, both severity 3.
However, it is observed that the effect of a void, usually within a small area, from both structural and aerodynamic perspectives,
is not comparable to the effect of larger damage, such as peeling, chipping, or erosion. The weighing for each defect type was
introduced to balance the score of a blade with many minor defects compared to a blade with a few large ones, which are more
likely to be repaired.

In the weighing scheme, each defect is given a score from 0 to 1, based on how likely it is to motivate a repair on the blade. A
stand-alone peeling or chipping of severity 3 will not be enough for a repair, however, several of these defects exposing surface
laminate would be repaired to prevent structural erosion and increased cost. It is standard practice to repair erosion just before
it starts to affect structural laminate.

Severity 4 erosion will be a cause for a repair in 100 % of the properly maintained wind farms, so we have selected it to be
the highest end of the scale. On the other hand, a severity 1 void will not motivate a repair and is often seen in the very early
life of the turbine, as it is a result of manufacturing imperfections. That is the reason it is given a very low score on the scale,
as you would expect the presence of multiple minor voids on the majority of blades in operation. These voids are not going to
lead to a repair on their own.

Chipping and peeling are usually intermediate steps in the erosion development, based on the failure mechanism of the
coating. Some coatings will fail more homogeneously, in which case erosion progressing from severity 1 to severity 3/4 will
be observed, gradually increasing in length from tip to root. In other cases, the failure modes will be more centered around
pre-existing manufacturing defects, causing the defect evolution to be voids, chipping, erosion, or peeling to erosion. In this
case, there will not always be distinctive erosion development, so a combination of chipping and peeling defects exposing
surface laminate can be the cause for a repair.

For the purpose of modeling defects related to leading edge erosion, only defects observed on the leading edge of the
blades are included in the analysis. It is assumed that only the defects specified in Figure 5 are caused by erosive impacts. For





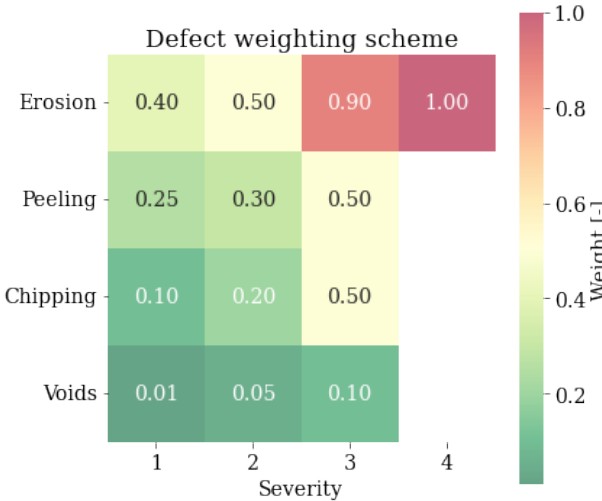

**Figure 5.** Scheme that is being used to assign weights to each defect. The weights represent the urgency of repairs where weights close to zero are not non-structural defects and weights close to one are threatening the structural integrity of the blade.

that reason, other types of defects such as gouges, scratches, or cracks are not included in the analysis. Finally, it should be

mentioned that certain blade inspections were deemed invalid because of uncertainties related to inspection quality or possible unknown repairs and the affected blades have therefore also been removed from the analysis. After performing such filtering, a total of 8501 defects were included for the overall analysis, distributed across 562 different wind turbine blades. The marginal distribution of all defects is visualized in Figure 6. It shows that more than 70 % of all the observed defects are categorized as voids, where the majority of them are severity 1. These defects are considered non-structural and do not pose a risk to the

blade integrity. However, it is expected that these defects can accelerate the progression to more severe defect types over time. Only a very small portion of all the observed defects are categorized with a weight of 0.5 or above.

Based on the blade inspections, the physical distribution of defects along the blade is visualized in Figure 7. We observe that the bulk of the defects are located on the outer 60 % of the blade but not significantly more on the region of the tip. However, looking at the mean defect weight along the blade, we observe a clear trend where the most critical defects are located in the

tip region. Generally, we also observe increasing variation in defect types and severities as we progress towards the blade tip.

The type and quality of blade inspections are not subjected to any official standards and are rather based on the urgency and available resources in terms of workers, equipment, etc., at the time of the inspection. This complicates the comparison between inspections from different sites and it is, therefore, necessary to define a robust methodology for making a fair comparison that also reflects the physical progression of the erosion damage. In addition, we are typically challenged by the scarcity of blade

inspections in combination with operational and weather data. In this study, we will use a simple encoding strategy that takes basis and benefits from the way repair recommendations are typically made. While the full blade inspections provide valuable information about all the defects observed on a blade, it is often the most critical defects that determine whether a repair is





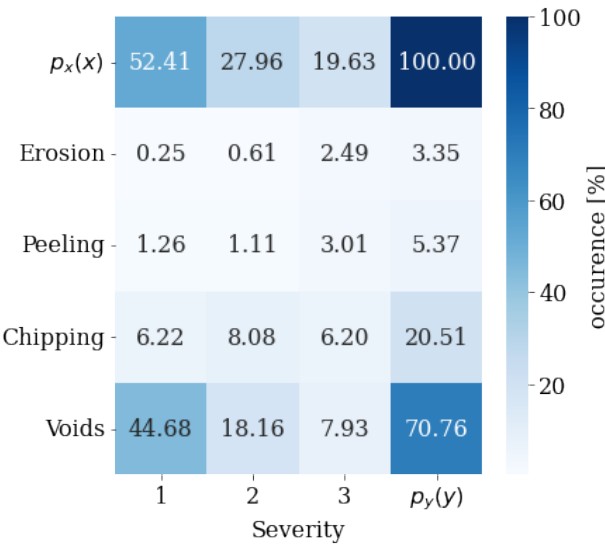

**Figure 6.** Marginal distribution of all observed defects after basic filtering.

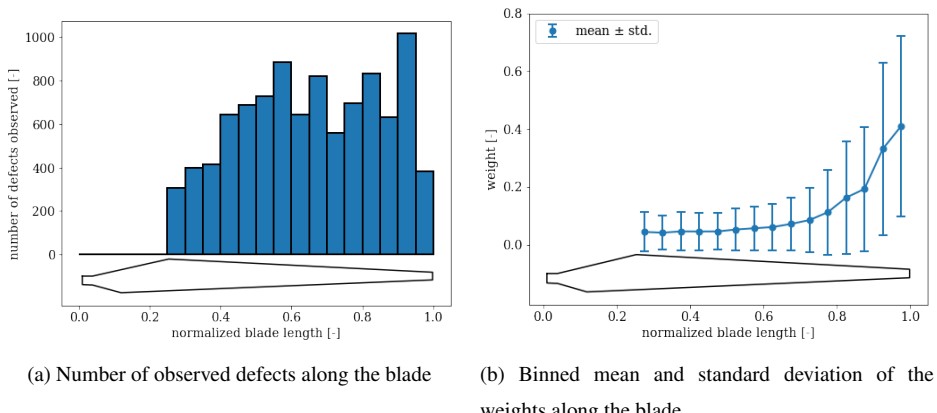

(a) Number of observed defects along the blade

(b) Binned mean and standard deviation of the weights along the blade

**Figure 7.** The total number of defects observed per cross-section of the blade (left), and the corresponding defect weights (mean and standard deviation, right).

recommended. For each inspection, we loop through the individual inspected blades and extract the most critical defect with respect to the weighting scheme. By taking the average weight of the most critical defects across the entire wind farm, we get

an unbiased point value that represents the overall erosion damage state of the wind farm at the time of the inspection. Also, we get a distribution of the most critical defects which indicates the defect variability within a wind farm. Such information can be used to post-process model predictions, thereby decoding the damage state back to the well-defined defect types and severities from Figure 5. In addition, we can then use available information about the time of inspections and/or the commissioning





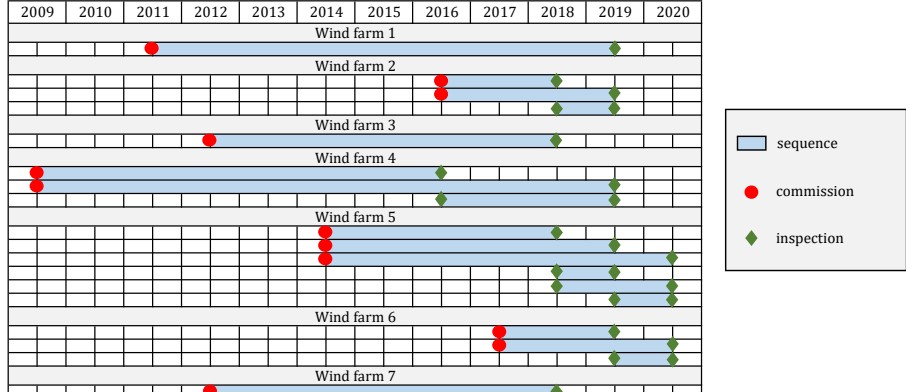

**Figure 8.** Visualization of all the available data sequences used for generating time-aggregated input and target features. The sequences are shown by light blue blocks and can either start from commission (orange dots) and end with an inspection (green diamonds) or they can start from inspections and also end with inspections.

date to map a sequence of weather data to our encoded damage state. This allows us to compute the damage progression, i.e.,

the difference between the damage observed at the time of inspection and the initial damage observed at the beginning of a sequence. Naturally, the initial damage is assumed to be zero when the start of a sequence is the commissioning date. However, for wind farms where two or more inspections have been performed, we can generate samples between two inspections where the initial damage is not zero but instead, the damage observed from the previous inspection. We use this information to generate an additional feature, the initial damage, which can be used as an input to the prediction model. The application of the

described sequential transformation is visualized in Figure 8, which shows all the available sequences for the seven wind farms used in the study. For each sequence, the start and end are indicated with a marker. Red dots indicate the time of commission and green diamonds indicate the time of inspection. As seen, the majority of the data are based on commission-to-inspection sequences while some are based on inspection-to-inspection sequences. Essentially, we aggregate sequences of weather data to map the encoded damage state observed at the end of that sequence.

**2.3 Data-driven modeling**

The commonly known curse of dimensionality refers to the rapid increase in variable space volume when the dimensionality increases. Working with blade inspections that are already expensive and time-consuming to collect, sparsity becomes a real issue, and it is key to identify and select appropriate features to be used as inputs for a data-driven model. Not only does adequate feature selection improve reliability, but it also allows for better interpretation of the model results and combat feature

redundancy. It has become common knowledge that leading edge erosion is a multidisciplinary process that is potentially affected by a multitude of different variables such as meteorological conditions, blade properties, and operational characteristics of the wind turbine(s) (Prieto and Karlsson, 2021; Tilg et al., 2021). Due to sparsity and limitations in data availability with



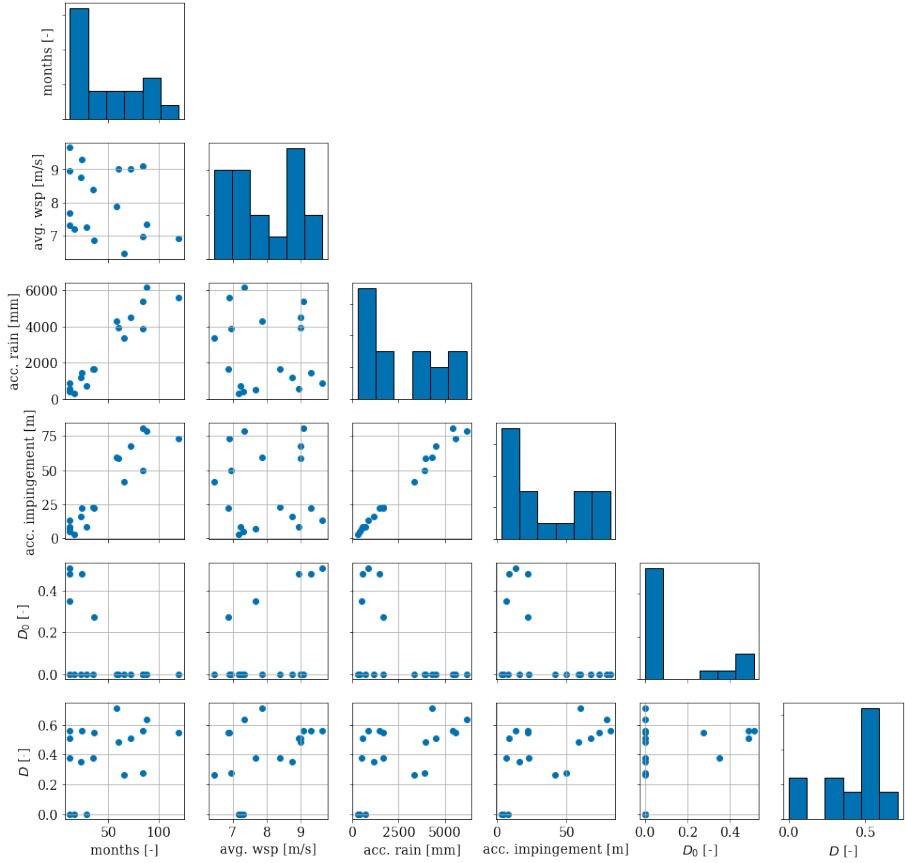

**Figure 9.** Correlation matrix showing the pairwise relationships for key parameters in the dataset as well as histograms in the diagonal. $D$ refers to the encoded damage at the end of a sequence (used as target feature), $D_0$ is the encoded initial damage observed at the beginning of a sequence (used as input feature), avg. wind speed, acc. precipitation, and acc. impingement (used as input feature) refer to the average wind speed, accumulated precipitation, and accumulated impingement, respectively, for the available sequences.

regard to the mesoscale weather data and operational SCADA data, a pragmatic engineering approach is taken in the present study where the accumulated rain impingement is the only meteorological input to the model.

Using the accumulated rain impingement as input has different advantages. Firstly, it should be stressed that the combination of rain and high wind speed (high impact velocity) is the main contributor to erosion damages and it is, therefore, critical to use an input feature that introduces this combination. Strong wind without any rain or heavy rain without any wind, will, theoretically, impose little to no erosion damage on a wind turbine blade. Typically, the accumulated rain can be used as a very good proxy which can also be seen in Figure 9 which shows the pairwise relationship in the dataset. The accumulated rain

and the accumulated impingement are highly correlated but essentially there is no guarantee that the impingement is high just because the accumulated rain is high. This emphasizes the main advantage of using the accumulated rain impingement as an input feature. A similar approach is used for other lifetime surface models previously in the literature, e.g., Bech et al. (2022),





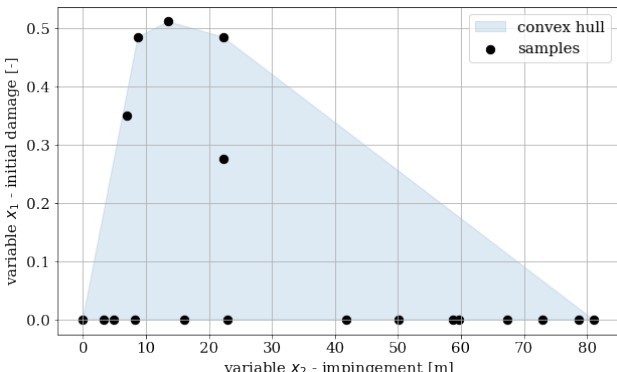

**Figure 10.** Visualization of the two-dimensional input variable space covered by the available training data and the corresponding convex hull.

and it was concluded that the rain impingement is a good global measure for quantifying the impinging rain on wind turbine blades. Secondly, as the rain impingement combines wind and rain in a single feature it acts as an inherent dimensionality reducer. For the prediction of leading edge erosion damage, where blade inspections are indeed very sparse, a limited number of input features is desired. Figure 10 visualizes the two-dimensional input variable space covered by the available dataset. The convex hull is indicated by the shaded area and it is observed that the samples are grouped in two areas. One group covers the variable space with no initial erosion damage with the samples being roughly equally distributed. The second group is smaller and covers the variable space with a limited impingement range. The already limited variable space coverage supports the choice of using as few input features for the modeling problem. The probability of a sample falling within the convex hull decreases for higher-dimensional problems. For this reason, using more input features would simply require a larger dataset. It should be mentioned, that several different combinations of input features were tested and evaluated. In the end, it was found that by introducing more features, the explainability of the model was reduced even if the statistical performance was slightly improved. When working with such few training samples, choosing the optimal input features becomes a trade-off between accuracy and interpretability.

It is a general understanding that ML requires a high amount of data (both in terms of its size and variety within the domain it covers) to be applicable and robust. While that is true for certain ML applications, there are many types of problems where simpler data-driven algorithms are useful, even with few training samples. Generally, the specific requirements for data size are ambiguous and depend on the complexity of the problem that is being mapped and the complexity of the chosen model architecture. For the purpose of best utilizing the available number of samples for our particular modeling problem, we will implement exhaustive bootstrap aggregation by using a leave-$p$-out cross-validation technique (Breiman, 1996). This technique allows for creating an unbiased ensemble of datasets based on all the available number of samples. Considering a dataset of size $n$, the technique consist in taking out $p$ samples for testing and then training on the remaining $(n-p)$ samples. This procedure is then repeated for all unique combinations. The total number of unique combinations can be calculated as:





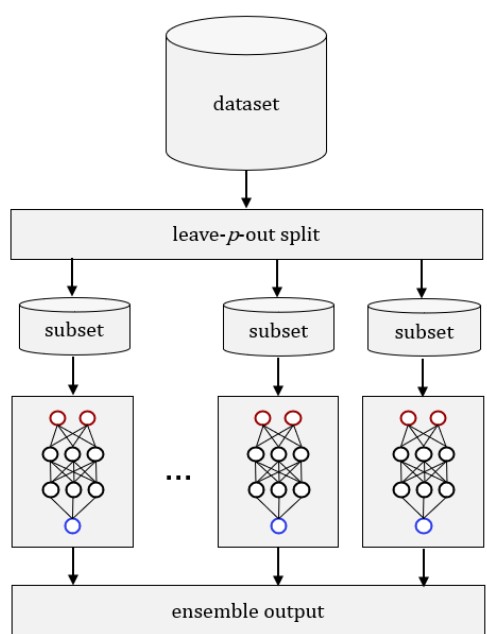

**Figure 11.** Visualization of the ensemble modeling concept using a leave-$p$-out technique for generating subsets.

$$\frac{n!}{p!\,(n-p)!} \qquad (4)$$

As indicated above, this technique is not very feasible for large sample sizes as the minimum number of unique combinations always will be equal to or larger than the original number of samples and drastically increase as $p$ increases. In our particular case where only a limited amount of data is available in terms of target values, the technique is very beneficial and allows for creating an ensemble of subsets. Each subset is used to train a single model, often named a weak learner. The individual models

are collected in an ensemble which allows to make ensemble predictions from where statistical properties such as variance and expected value can be estimated (Lakshminarayanan et al., 2017). A flow diagram of the ensemble model concept can be found in Figure 11.

It is necessary to evaluate the optimal splitting strategy as there is a trade-off between bias, variance, and computational time. Here in this study, we evaluated 16/2, 15/3, and 14/4 split configurations, corresponding to a testing portion of 11%,

17%, and 22%, respectively. It was found that the 15/3 split was best suited for the task, and ultimately resulted in an ensemble consisting of 816 individual weak learners each trained on a unique dataset.





One issue of modeling a problem with a limited amount of training data is the ability to validate the performance of the model. Combining data-driven and physics-based model validation, here we performed physicality checks which provide a qualitative assessment of the model output over a representative variable space. This is done to ensure that the model outputs

comply with the known physical behavior of erosion. Criteria for physical compliance include:

– Only damage progression is allowed, i.e., the change of damage over time must not be negative

– Model results should have a zero-intercept

– When the impingement is zero, the predicted damage should be equal to the initial damage state

– The incubation period for the damage progression should be captured by the model

To accommodate the need for physical interpretability, synthetic data were introduced at one of the boundaries of the variable space, namely the zero-impingement boundary where the response is well-known (i.e., zero rain – zero damage progression). The synthetic data forces the model to comply with the physics and it allows for better interpolation and variance reduction.

It was chosen to use a simple feedforward neural network (FFNN) as the weak learners for the ensemble model. There are several arguments for why neural networks are preferred. First of all, neural networks have the ability to learn non-linear

relationships. In addition, they are able to interpolate and, to some extent, extrapolate, which is not the case for other machine learning classes such as support vector machines or those based on decision trees. At the same time, simple neural networks are also well suited as weak learners for ensemble modeling, whereas simple linear regression models are not. Several of these other models were evaluated and tested doing the model evaluation process. Specifically, it can be mentioned that polynomial chaos expansion (PCE) up to third order was evaluated in an exhaustive manner, i.e., by evaluating all the polynomial expansions

repetitively. It was found that a first-order model performed best whereas higher-order models failed to represent the physicality requirements specified earlier. However, neither were found to outperform the ensemble performance from simple FFNNs.

The optimal hyperparameters were determined using an exhaustive grid search over a reasonable parameter space. The final architecture of the neural network consisted of two layers, each with 5 neurons. No performance improvement was found when introducing regularization in the form of dropout or L1 and L2 regularization. The weights were initialized by $\mathcal{N}(0.5, 0.5)$

and the biases were initialized by zeros. The relu activation function was used for all layers. The model was trained using the Adam optimizer with a learning rate of 0.01 and the mean squared error as the loss function. The models were trained for 5000 epochs with an implementation of early stopping if no improvement was observed through 200 consecutive epochs. It has been suggested by Naftaly et al. (1997) to not implement early stopping in ensemble learning as the overall variance is theoretically reduced with no impact on the overall bias. However, it was found from validating both training methods that the

overall variance reduction could not be justified based on the limited sample size available for training. When applying early stopping, the ensemble variance increased but the average prediction error was reduced. The opposite was found when no early stopping was implemented, and the trade-off between variance and bias was chosen to be in the favor of the overall bias.





# 3   Results

## 3.1   Training validation

Training an ensemble of models using a leave-$p$-out technique has the benefit that each sample from the original dataset is predicted by several independent and unbiased models. This allows for evaluating the performance of the individual models, but more importantly, it allows for assessing the distribution of predictions from the ensemble. The characteristics of the distribution give an indication of the model uncertainty, whether that be noise in the data or model uncertainty caused by insufficient data, hindering the model to learn. Figure 12 shows a direct comparison between the ensemble model predictions

and the 18 observations of the true encoded damage. Using a 3/15 split, each sample is predicted by 136 individual models. The dots show the ensemble average of each sample and the error bars indicate $\pm$ one standard deviation around that mean. Generally, the validation shows a good comparison between mean predictions and observations which is supported by the error statistics also shown in the figure. The fitted trend line (red dashed) indicates a general underestimation of damages above 0.25 of 5–7 %. However, we do observe a few samples being significantly underestimated by the ensemble mean. From the

error bars, we observe a very small ensemble variability for samples with low damage, whereas samples with higher damage, generally have an increased ensemble variability indicating less model certainty. The ability to estimate the model uncertainty is a great advantage for the overall result interpretation but also for the applicability.

The ensemble error distribution is visualized in Figure 13. It is based on a total of 2448 error values equally distributed on the 18 samples. The mean bias is seen to be very close to zero, where the error distribution is observed to have a slightly

bimodal appearance with a secondary peak between -0.2 and -0.1. This corresponds to the general underestimation previously described. With such a limited data size for training, the importance of individual samples increases, and the few consistently underestimated samples reoccur in many of the estimators thereby causing the second peak. In addition, we observe a significant amount of errors outside of the interquartile range (IQR) which is not uncommon in ensemble learning where the focus is on the ensemble prediction rather than predictions from the individual learners.

Finally, we validate the ensemble model through physicality checks by qualitatively evaluating the ensemble output over a representative variable space. Figure 14 shows the ensemble mean (left) and standard deviation (right) for different combinations of the two input features. Looking at the ensemble mean, we quickly validate the previously defined criteria of zero damage at the intercept and damage progression as the input features increase. Following the one-dimensional slice for a zero initial damage, we identify what can be characterized as the incubation period. Typically, the erosion progression is separated

into three periods; incubation, transition, and steady-state (Springer et al., 1974). Though we also see this characteristic from Figure 14, it should be noted that the encoded damage used in the present study, represents an overall damage state. For this reason, it can only to some extent be compared directly to the damage progression observed from experimental erosion testing.

The ensemble standard deviation gives an indication of the certainty of the model across the variable space. The overall certainty is heavily restricted by the low number of samples but we still observe local areas in the variable space where the

individual learners predict similarly. Specifically, this involves predictions at the zero-intercept and around the boundary of the initial damage. These areas were already restricted by the artificial samples and we, therefore, expect less uncertainty here.

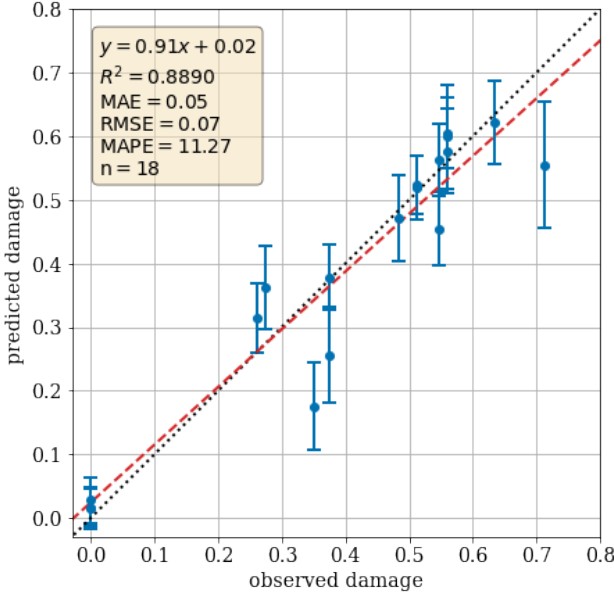

**Figure 12.** One-to-one comparison of ensemble predictions and the true observed damage (both encoded). The error bars indicate the ensemble standard deviation, the red dashed line shows the best linear fit, and the black dotted line shows the results of a perfect prediction as a reference.

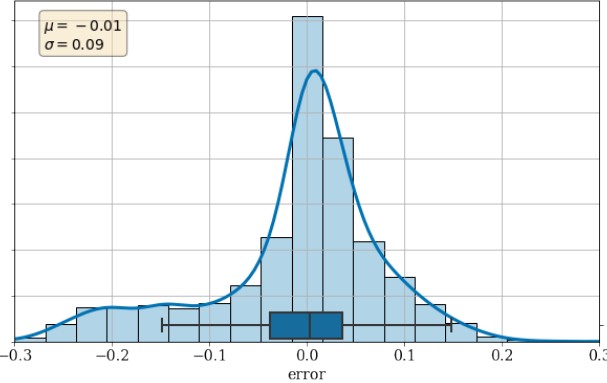

**Figure 13.** Error distribution of the 2448 predictions based on all subsets, i.e., each sample is predicted by multiple individual models.

Naturally, we also observe less uncertainty in the areas covered by the training samples as previously mentioned. It should be mentioned, that the ensemble standard deviation only describes the inter-variability of the ensemble and not directly the prediction uncertainty.

In addition to the validation of the response surface shown in Figure 14, we also validate the predicted damage for different sequence lengths. An example of the ensemble-predicted damage for three different wind farm sites is visualized in Figure 15,

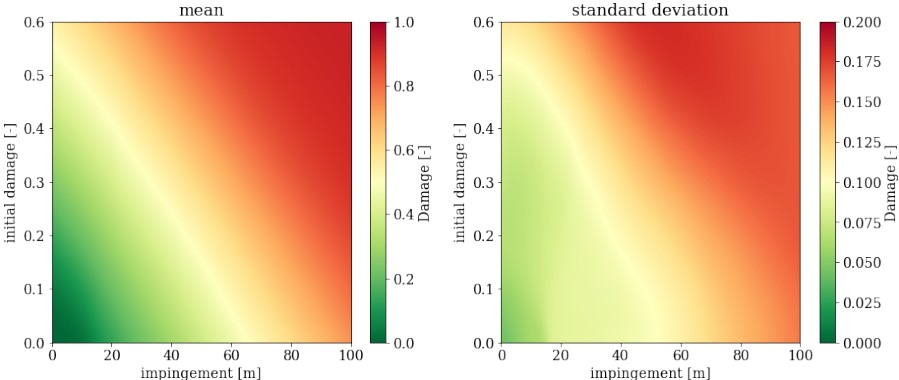

**Figure 14.** Color maps used as physicality checks to validate the ensemble model response across the two-dimensional input space. The left figure shows the ensemble mean and the right figure shows the ensemble standard deviation.

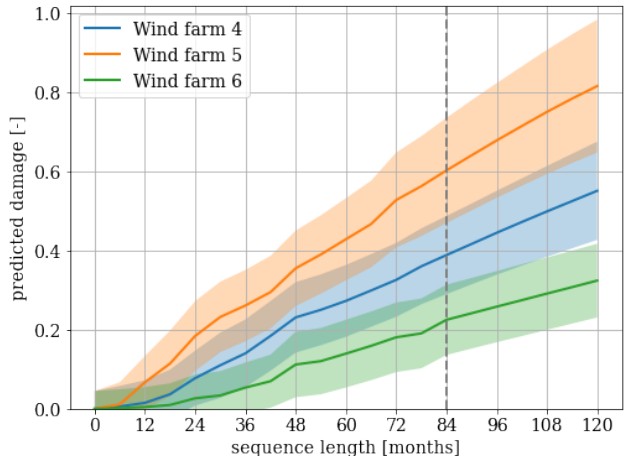

**Figure 15.** Example of the predicted damage progression for three different wind farms for varying sequence lengths and zero initial damage. Each sequence begins on 01-01-2013.

where the sequences always start on 01-01-2013. This is done to demonstrate the capability of both historical data-based estimations and forecasts based on climatological characteristics using a simple measure-correlate-predict approach. As mentioned, the weather data are only available in the period from May 2013 to January 2021, which means all estimates after 84

months are extrapolations. The cut-off point between prediction and forecasting is indicated by the gray dashed line. This is also observed by the linear appearance after the cut-off, compared to the non-linear behavior before. Figure 15 also shows a clear difference in damage progression between the three sites. One site is expected to experience erosion damages twice as fast as the other. While this is simply an example of three test sites, it clearly emphasizes the requirement of site-specific repair and maintenance planning.





## 3.2 Erosion map

Having trained and validated the erosion prediction model, we are able to use it to make time-specific predictions of erosion damages based on historic reanalysis data or use statistical climatological characteristics to make forecasts. Figure 16 visualizes erosion forecasts at different test sites across North Europe for three different cases. For all cases, the operational characteristics of the wind turbine are assumed to be that of a Vestas V80-2000 (see Table 1 for specific values). The first case is simulated for an operational period of one year with zero initial damage, the second case is simulated for an operational period of five years with zero initial damage and the third case is simulated for an operational period of two years with an initial damage of 0.4. The forecasts are based on statistical climatological characteristics obtained from eight years (2013-2020) of mesoscale data. The left column shows the ensemble mean and the right column shows the ensemble standard deviation.

After just a single year of operation, we observe that almost all sites are expected to have zero or very low erosion damage meaning, the erosion model is able to partially capture the incubation period. We also observe the ensemble standard deviation to be relatively low for all sites, indicating that the individual models agree well on the ensemble mean. Looking at the second case, i.e., after five years of operation, we observe a much different erosion map and clearly identify several sites that are expected to have severe erosion defects. Especially in the region along the southwest coast of Norway, we observe all three sites to have significant erosion damages after five years of operation. Using the weighting scheme in Figure 5 as a reference, we would expect wind farms in these regions to have major repair campaigns performed within the first five years of operation. As reported by Lussana et al. (2018), these regions are known to have the highest amount of annual precipitation and as the accumulated impingement is very closely related, we naturally expect to see this behavior from the erosion model. Similarly, we also observe distinct sites in the United Kingdom, Ireland, and the Faroe Islands which are exposed to more erosive conditions, highlighting the importance of the site-specific weather conditions. The erosion map also visualizes a general trend of coastal regions being more exposed to environmental conditions that cause erosion compared to more inland sites. While such findings have been reported in previous studies (Hasager et al., 2021; Verma et al., 2021b; Herring et al., 2020), it still provides valuable knowledge that is directly validated by actual blade inspections. From the map of the ensemble standard deviation, we also see how the ensemble model uncertainty increases for sites with high expected erosion damages, e.g., the three sites in the west of Norway. These findings are supported by the physicality validation from Figure 14, and inputs from the industry that performed inspection and maintenance around a number of these sites.

Finally, we also demonstrate the erosion model's ability to forecast erosion damages when a site is assumed to already have some initial, arbitrary damage. This is especially useful in the case where a wind farm has been inspected and the end-user (typically the owner/operator) is interested in knowing the damage progression based on the current status. For the given example, the initial damage is 0.4 corresponding to very mild erosion. After just two years of operation, it can be seen how many sites are expected to experience a noticeable damage progression. Naturally, the same sites that were previously explained to be exposed to harsh erosive weather conditions, are also the same sites that will experience a faster damage progression when having initial damage. The ensemble uncertainty is observed to generally be higher when forecasting with an initial damage. Similar to the analysis with zero initial damage, higher uncertainties are related to the available training data not covering the



variable space extensively, which does cause a higher variability among the individual learners. Comparing Figure 16(f) of the ensemble standard deviation to that in Figure 16(b) with no initial damage, significant differences are observed for a difference in operation period of just one year.

It should be mentioned, that the capabilities of the erosion model extend beyond what has been demonstrated here. The tool is generic and very flexible which allows the user to model different scenarios. This includes site-specific modeling of different wind turbine types with different rotor speed curves and/or rotor diameters and hub heights.

## 4    Discussion

Developing erosion models is indeed a challenging task as it involves many different disciplines. As mentioned in Section 1, existing erosion models often rely on accurate information about turbine operation, precipitation, and blade coating properties. While this type of engineering model offers a theoretical foundation for estimating erosion, such an approach also entails several assumptions and uncertainties which are difficult to justify and validate without field data. This fact was a major motivation in the present study, for developing a purely data-driven approach for modeling site-specific erosion damages based on mesoscale weather data and real blade inspections. However, this type of methodology also introduces new assumptions and uncertainties that need justification. One of the first problems that arise when working with any type of data, is the uncertainties related to data quality and integrity. For this study, the data can be separated into two categories, namely weather data and blade inspections. The weather data comes from a mesoscale NWP model and as with any model-simulated data, there will be some uncertainty and statistical errors. While the data provider, DMI, has performed internal, continuous correction and validation using conventional observation data, uncertainties remain and will propagate through any model (Yang et al., 2012; Nielsen et al., 2010). In terms of ML application, biases in the input data are usually not considered a problem as long they are systematic (Mehrabi et al., 2021). This is, however, not the case for analytical erosion models which rely heavily on the physical quantities of the input variables.

Another prominent issue with weather data from NWP models is the uncertainties related to the spatial and temporal resolution. Field measurements give an accurate estimate of the conditions at a distinct location, whereas mesoscale data represents a full grid cell. While this representation might be adequate for small grid cells with uniform land cover, the atmospheric variability over areas with complex terrain might not be very well captured. Considering the horizontal resolution of 2.5 km, which is relatively high for NWP model simulations, it is still expected that there will be some intra-grid variability that cannot be accounted for. This is especially the case for precipitation data which is notoriously known for being very local and highly time-varying (Letson et al., 2020a). Furthermore, conditionally unstable atmospheric conditions that can lead to moderately strong convective showers in mid-latitude climates are chaotic in nature (Lorenz, 1963). This can result in displacements of these showers by 10s of kilometers, even in good NWP forecasts.

Tilg et al. (2020) examined the potential for using vertically pointing radars to nowcast vertical precipitation profile. The purpose was to assess the feasibility of using such radars for erosion-safe mode operation and the study emphasizes the need for accurate precipitating data with a high temporal and spatial resolution. A time series of rain intensity from the study depicts

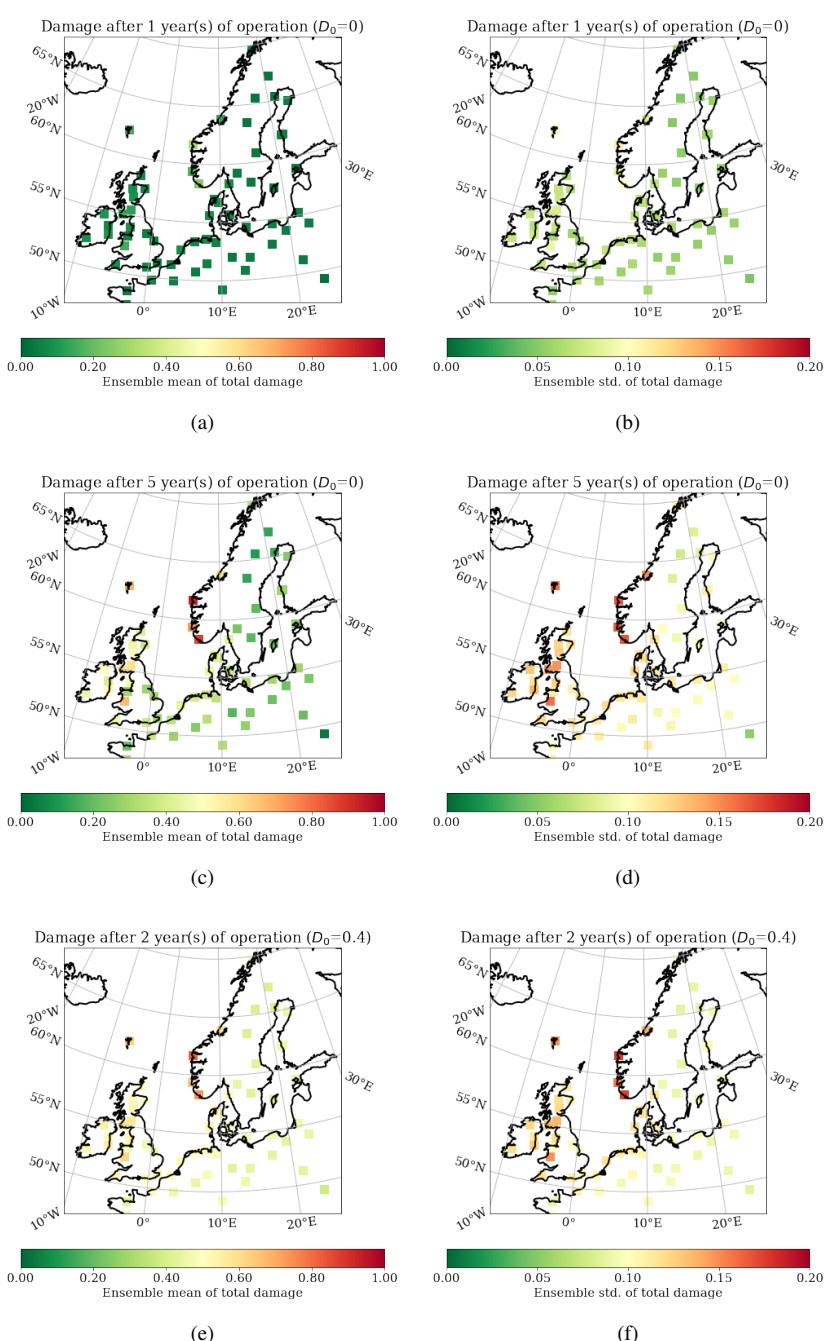

**Figure 16.** Erosion maps created for three different scenarios to demonstrate the versatile capabilities of the developed prediction tool. The left column visualizes ensemble means and the right column visualizes ensemble standard deviations for the three cases.





the high variability of precipitation within the time frame of a single hour. Considering this behavior, it could be argued that a temporal resolution of one hour is simply not high enough to provide adequate information about the rain characteristics that are needed for predicting erosion damages. Especially the inter-hourly periods with very heavy rain are impossible to account
for. While former studies have suggested that these rain intense periods contribute supremely to the erosion damage progression, the findings from the present study indicate that this contribution might not be as prominent. A small parametric analysis showed that the correlation between erosion damage progression and accumulated impingement decreased when applying a minimum rain threshold to the weather data. It indicates the importance of including low-intensity rain when calculating the impingement. Though the database is too limited to make any finite conclusion on this matter, it does give an indication that the
high occurrence of light rain also contributes significantly to the overall erosion damage. A similar observation was adduced by Herring et al. (2020) who suggested that erosion damage might not be driven solely by heavy and violent precipitation. This disagreement in literature could stem from the difference between erosion assessment from controlled conditions, e.g., rain erosion test where the equivalent to low-intensity rain is often not considered, and field observations where the accumulated impingement mainly is constituted of low-intensity rain. In addition, it should be mentioned that different precipitation condi-
tions might contribute differently to each of the stages in the erosion progression. As an example, mild rain might contribute more to the damage progression but the initiation of the damages after the incubation period is more likely to be driven by heavy rain events. While it has not been possible to validate this statement through decomposition of the erosion damages by rain intensity, we do observe an accelerated erosion damage progression when initial damage is present (cf. Figure Figure 16).

In terms of wind speed, we also expect the mesoscale data to feature certain ambiguity compared to actual observations at
465 the site. The computed impingement is dependent on the tip speed, and hence the incoming wind speed, which means that errors in wind speed directly affect the predicted erosion damage. One of the possible errors is related to the site-specific wind speed biases. If it is expected that the positioning of a wind farm is done optimally, with respect to the wind resource, it is not unlikely that the inter-grid wind speed is higher at the exact location of the wind farm. This is specifically the case in complex terrain where a high wind resource variability is to be expected, even for a single grid cell. This effect is less likely
to be present for offshore sites or for sites located in a grid cell with relatively uniform topography. The overall effect related to the ML-based erosion model presented in this study would be a general underestimation of the damage estimates. It should be mentioned that while it has not been possible to validate this claim based on the individual inspections, we do observe a small negative bias for ensemble performance, as described earlier. However, it is assumed that the wind speed uncertainty is generally not as large as the precipitation uncertainty. This is mainly caused by the difference in variability when considering
the given spatial and temporal resolution.

In addition, wake effects within the wind farms could potentially cause variation in erosion conditions. It is expected that wind turbines positioned in the front rows, generally will be exposed to higher wind speeds compared to wind turbines positioned deeper inside the wind farm. While flow variations across a wind farm due to wake effects are very well documented, the effect that it might have on the erosion condition is not. Though this is beyond our scope, the blade inspections used in the
480 present study have not indicated any noticeable variations.



We generally observe a lack of field-based blade inspections to validate existing erosion models in natural exposure. While the erosion model in the present study is founded on the basis of blade inspections, it also illustrates new challenges related to working with such data. First of all, one needs to accept the stochastic process that is leading edge erosion. This type of behavior is expected to be present when comparing individual turbines within the same farm and can partly be explained by operational differences. Individual turbines are for example expected to experience standstill periods for various reasons, whether due to failures or deliberate routines such as maintenance or power curtailment. These standstill periods change the environmental exposure for the individual turbines and therefore also the expected erosion damages. Operational data, which is typically obtained through the SCADA system, has not been available and downtime has simply not been considered when computing the accumulated impingement. As previously mentioned, the tip speed is directly estimated based on the assumed operational characteristics from Table 1 and this simplistic approach is based on the assumption that all turbines operate with a rotational speed given by a linear threshold function. It should also be stressed that no coating properties have been taken into account in the presented model framework.

In addition to the variance that inevitably will occur between individual turbines, we also need to acknowledge the inherent variability of erosion progression on individual blades. A qualitative comparison was performed for three individual blades on the same turbine. In this case, it must be assumed that the three blades have experienced almost identical operational conditions, both in terms of rotational speed and precipitation exposure. However, when assessing the defect distribution and characteristics from three blades on the same turbine, we do still observe notable differences. This inter-turbine variability was a recurrent behavior that was observed for many of the inspected turbines and underlines the stochastic process of erosion development and progression. While many studies attempt to model this very complex, and as proven, to some degree stochastic, erosion development, the approach used in the present study relies on a robust encoding scheme that is applied to entire wind farms thereby lowering the influence of inter-turbine variability.

In addition, there was a significant difference in the observed number of defects per blade between the individual inspections. As an example, for one inspection, the total number of defects per inspected blade was 146 whereas this number was only 1.15 for another inspection. Similarly, this difference was observed even for inspections from the same wind farm. Considering wind farm 5 (see Figure 8), where three comprehensive inspections were performed over three consecutive years. It was found that the total number of observed defects per blade decreased by 66 % from 2018 to 2019 and again by 46 % from 2019 to 2020. This difference can partly be explained by two things; firstly, the blade inspections used for this study were obtained using different methods. Generally, there exist three different types of inspection methods, namely drone-based inspection, ground-based inspection, and rope inspections. All three methods are acknowledged and provide the required integrity in terms of defect assessment. However, they might differ in the way post-processing and reporting are performed which can cause differences in the observed number of defects. Secondly, as the overall condition of a blade goes from mild defects to more critical defects, the focus of the blade inspection will be on the most critical defects that potentially could require repairing. For this reason, we also expect the number of defects per blade to be highest for wind farms where the overall damage is not acute. While this trend at first glance might appear to be unfavorable, it actually confirmed the advantage of using the encoding strategy presented in this study. Using an average of the per-blade defect with the highest weight, allowing for a robust and fair





comparison between inspections, which ultimately was the goal of the encoding. From a qualitative assessment, this was also validated by confirming that the encoded damage did indeed progress realistically over time for the wind farms where multiple inspections had been conducted. In addition, we also did show in Figure 12 that the encoded damage was predictable by the input features, thereby also validating the encoding strategy.

We have proposed using a simple feedforward neural network architecture as the weak learner in our ensemble model. The purpose of this approach is to best utilize the very limited sample size that is available for training and testing. In addition, the ensemble learning allows for generating mean and variance estimates of the individual learners. This helps reduce the generalization error through a bias-variance trade-off. While a theoretical framework has been proposed by Bonab and Can (2016) for determining the optimal number of individual learners, the splitting strategy used in the present study was chosen

based on a qualitative assessment of three different splits. This approach is very useful in the present study, but it might be impractical in cases where more blade inspections are available. In such a case, other splitting techniques could be evaluated to find the most suitable.

    The ensemble model presented in this study takes only two variables as inputs, namely the initial damage at the start of a sequence and the rain impingement accumulated throughout the sequence. These two descriptive variables were found to be the

best suited for mapping the target feature. As mentioned, the accumulated rain impingement captures both rain and wind speed in a single feature. Generally, rain is assumed to be the main contributor to erosion damages on wind turbine blades, mainly because of the higher frequency at which rain occurs compared to e.g., hail or graupel. While this is true for the spatial domain investigated in the present study, other locations might be exposed to other precipitation conditions (or other airborne particles) that contribute significantly to the overall erosion damage (Punge and Kunz, 2016; Prein and Holland, 2018). In addition to the

generally larger size of hail compared to rain, it must have a larger impact than a rain droplet or a snow crystal since these will easily be deformed or broken when hit. The hail particle on the other hand is a solid quasi-sphere of ice that is much more rigid. For that reason, hail should be included in more detail as meteorological input data for future predictive erosion studies. As an example, radar measurements from several states in the United States of America, have shown substantial differences in hail frequency and severe hail events, indicating the importance of assessing site-specific erosion drivers (Letson et al., 2020b). In

addition, several other environmental parameters might influence the rate of erosion. This includes lightning, rapid temperature variations, icing, sea spray, UV radiation, etc., but the effects have not been documented in the literature. While model data to account for these conditions have not been available in the present study, the effects are also expected to be extremely difficult to verify from the relatively few blade inspections available, especially considering the constricted domain offering limited variation for the aforementioned parameters.

**5   Conclusions**

Wind turbine blades are operating with tremendous tip speeds and small particles such as rain droplets will have an erosive effect on the leading edge throughout the lifetime of the wind turbine. While this phenomenon is well known, the challenge of modeling leading edge erosion arises from the highly multivariate, complex, and to some extent stochastic process that erosion





is. In addition, the sparse extent of quality blade inspections makes it extremely difficult to validate existing engineering models
in real-life conditions.

In the present study, we have presented a data-driven framework for modeling blade defects from leading edge erosion on
wind turbine blades. The framework provides an erosion prediction tool that is based on a machine learning model trained
using mesoscale numerical weather prediction (NWP) and real blade inspections from several wind farms in Northern Europe.
The framework is two-folded and consists of a training and application phase.

The training process is governed by the available blade inspections and corresponding site-specific weather data. The blade
inspections are encoded using a defect weighting scheme to consider only the most critical defects. The weighting scheme
represents the urgency for repair actions and fits directly into the terminology used by the industry for making maintenance
planning and repair recommendations. The mesoscale weather data is used to estimate the site-specific rain impingement
and is time-aggregated to be compatible with the blade inspections. The preprocessed blade inspections and weather data are
forwarded to an ensemble learning algorithm that splits, trains, and validates the ensemble model. The ensemble model consists
of hundreds of weak learners in the form of simple feed-forward artificial neural networks and allows for estimating model
output statistics.

In the application phase, the trained model is used in combination with new weather data and user-defined wind turbine
operational characteristics, to make site-specific damage predictions and/or forecasts. The trained predictive model can be used
interactively to provide a live erosion map based on available weather data or to make site-specific erosion predictions for new
or existing wind farms.

Though the present study proposes a robust methodology for modeling leading edge erosion defects, it does rely on several
assumptions and uncertainties related to the weather data, operational conditions, and blade properties. For this reason, the
model should be used as a low-fidelity tool to support site-specific planning and scheduling of repairs as well as budgeting of
operation and maintenance costs.

*Code and data availability.* The mesoscale NWP model data will be available through DMI's Open Data API at the beginning of 2023
from https://confluence.govcloud.dk/display/FDAPI. The blade inspection data provided by Wind Power LAB (WPL) are confidential and
therefore not publicly available.

*Author contributions.* JV had the lead on paper writing, data analysis, and modeling. TG contributed to the data analysis and model devel-
opment. CBH contributed to the design of the overall project. HS and MH contributed to the blade inspections and KPN contributed to the
mesoscale weather data. All contributed to the writing of the paper.

*Competing interests.* The authors Hristo Shkalov and Morten Handberg are employed by the private company Wind Power LAB (WPL),
and the author Kristian Pagh Nielsen is employed by the Danish Meteorological Institute (DMI), administrated by the Danish Ministry of




Energy, Utilities and Climate. The authors declare that they have no other known competing financial interests or personal relationships that could have appeared to influence the work reported in this paper.

*Acknowledgements.* The authors would like to acknowledge the support from Wind Power LAB, DMI, and DTU who were involved in the "Blade Defect Forecasting" project. The good collaboration has contributed greatly to the development of this paper.

*Financial support.* This research was partly funded by the Innovation Fund Denmark Grand Solutions Grant 9067-00008B "Blade Defect Forecasting".





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
