# Peer review of "Introducing a data-driven approach to predict site-specific leading edge erosion from mesoscale weather simulations"

_Wind Energy Science, 2022_

## Author Comment (AC1)

**Response to Anonymous Referee #1 comments of Manuscript ID WES-2022-55 entitled "Introducing a data-driven approach to predict site-specific leading edge erosion"**

Thank you for taking the time to review our article. We have addressed your comments attentively, for which the details are provided below.

1. Define NEA and DKA domains more appropriately in the text.

   An additional description has been added, which reads as:

   "The DKA domain consists of an 800x600 model grid with 2.5 km grid spacing. Each model column in the grid has 65 vertical levels with hybrid coordinates that follow the terrain near the surface and fixed pressure levels at the top of model grid. The lowest model level is approximately 12 m above the surface and the highest level is at 10 hPa. The grid has a Lambert conformal conic projection with 25.0°E and 56.7°N as the respective reference longitude and latitude, and 8.2°E and 56.7°N as the central longitude and latitude. The NEA domain has a 1280x1080 model grid. It has the same grid spacing and vertical levels as the DKA domain. NEA has a Lambert conformal conic projection with 25.0°E and 60.0°N as the respective reference longitude and latitude, and 7.0°E and 60.0°N as the central longitude and latitude."

2. It is unclear the reason for the $\alpha$=0.16 adopted by the power law extrapolation. This extrapolation procedure should be further explained and, furthermore, the sensitivity of the measured quantities with respect to $\alpha$ should be investigated.

   First of all, the value used in the study was 0.143 and not 0.16 as stated in the manuscript. This has been corrected. Secondly, to answer your comment, as mentioned in Section 2.1, the mesoscale weather data was provided at different model level heights. For each site, the model level height closest to that of the corresponding hub height was used. On average, the absolute extrapolated distance (i.e., between model height and hub height) was 5.41 m and the maximum distance was 11 m. A sensitivity study was carried out and it was found that changes in the shear exponent had very small impact on the accumulated impingement which is heavily dominated by the rain rate. For shear exponents in the range of 0.143±0.5, the maximum change in accumulated impingement was 1.7%. Since this sensitivity was performed for consistent shear exponents, the specific value is considered negligible for our study. It should also be mentioned that for the application of the model, the user can easily extrapolate using different shear exponents or even other extrapolation models. This information is now included in Section 2.1 to make it clearer for the reader/user.

3. The main input to the data-driven model is the accumulated rain impingement. Despite it does combine the effect of amount of rain and wind speed, it does not consider a third important parameter which is the rotational speed of the wind turbine. In the paragraph related to line 260, please expand the discussion while taking this argument into account. What would be the implication of including the rotation speed into the prediction models?

   The accumulated rain impingement is actually calculated based on the rotational speed, $\omega$, as indicated explicitly in Eq. 3. The rotational speed directly affects the tip speed which is proportional to the rain impingement.

4. The authors discuss that the feedforward neural network with a 2-layer architecture with 5 neurons per layer + RELUs was enough and outperformed other methods such as PCE, support vector machine. The ability to learn non-linear relationships is not exclusive from FFNN, and the RELU effect to allow a linear piece-wise description of the neuron weights can be emulated by another model based on moving averages or any piecewise form of approximation, for instance. The authors are encouraged to improve all the discussion surrounding the choice for the FFNN, exposing also its main limitations and the results of the PCE that was largely investigated.

   The authors fully agree that the ability for FFNN to learn non-linear relations is not exclusive. The choice of model was based on the validation criteria described in the manuscript. A comparison of the error statistics is shown below: The section describing the model selection has been updated to include a better comparison and reads as:

   "It was chosen to use a simple feedforward neural network (FFNN) as the weak learners for the ensemble model. Several models were evaluated as candidates by means of the validation described previously, i.e., statistical performance and physical interpretability. The neural network was found to be the best model when comparing error statistics. As an example, the root-mean-square-error was found to be 30.0 % lower than the support-vector-machine, 36.4 % lower than polynomial chaos expansion (PCE) of first

| model | fit | R² | MAE | RMSE | MAPE | FFNN's relative [%] comparison | | | |
|---|---|---|---|---|---|---|---|---|---|
| | | | | | | R² | MAE | RMSE | MAPE |
| FFNN | 0.91x+0.02 | 0,8890 | 0,05 | 0,07 | 11,27 | - | - | - | - |
| Linear regression | 0.84x+0.05 | 0,7395 | 0,08 | 0,11 | 17,32 | 20,2 | -37,5 | -36,4 | -34,9 |
| Decision tree | 0.89x+0.01 | 0,7347 | 0,08 | 0,12 | 17,17 | 21,0 | -37,5 | -41,7 | -34,4 |
| KNN | 0.70x+0.03 | 0,6242 | 0,12 | 0,16 | 27,11 | 42,4 | -58,3 | -56,3 | -58,4 |
| PCE 1st order | 0.85x+0.04 | 0,7409 | 0,08 | 0,11 | 17,50 | 20,0 | -37,5 | -36,4 | -35,6 |
| PCE 2nd order | 0.89x+0.02 | 0,6961 | 0,09 | 0,13 | 19,33 | 27,7 | -44,4 | -46,2 | -41,7 |
| PCE 3rd order | 0.86x+0.03 | 0,6659 | 0,10 | 0,14 | 23,13 | 33,5 | -50,0 | -50,0 | -51,3 |
| SVM | 0.82x+0.05 | 0,8018 | 0,07 | 0,10 | 14,77 | 10,9 | -28,6 | -30,0 | -23,7 |

order (46.2 % for second order) and 41.7 % lower than decision trees. For other error metrics the same behavior was observed. In addition, the neural network was also found best suited for capturing physical characteristics such as the incubation period. Specifically, it can be mentioned that PCE up to third order was evaluated in an exhaustive manner, i.e., by evaluating all the polynomial expansions repetitively. It was found that a first-order model performed best whereas higher-order models failed to represent the physicality requirements specified earlier.

When using ensemble techniques such as bagging, it is desired to use a high-variance model which is able to learn non-linear relations. This is because the variance is reduced through bagging and the bias is increased (or maintained). This trade is not unique for neural networks and other models such as those based on decision trees or expansion models can have the same ability."

5. Have the authors considered the use of classical surrogate models such as the Kriging method?

Specifically, the Kriging method has not been considered as it only allows for interpolation within the convex hull whereas other regression techniques allow for extrapolation also. This inability would also cause errors in the ensemble training process as we here require extrapolation in some of the data splits when validating the against the test data. Simpler models, such as linear regression were also considered but outperformed by the FFNN. This information will be clarified in the description of the model selection and reads as:

"Finally, it can also be mentioned that the ensemble training process used in this study, requires extrapolation in some of the data splits when validating against the test data, and though extrapolation is never recommended for ML applications, the FFNN has ability to do it anyway and performed better than SVM or models based on decision trees. It is the inherent definition of decision trees that makes them unsuited for extrapolation, e.g., the output of a decision tree is limited by the leaf nodes and can therefore never exceed the outer leaf nodes."

6. The authors should present one application case that applies the developed framework to predict damage in a new site within the covered region, focusing mainly on the workflow that would need to be followed. As the caption in Figure 3 describes, the trained model is evaluated on 99 new sites across the region of interest. This is our example of an application of the developed framework, in this case shown as the erosion maps in Figure 16. There we show simulated cases for the sites that were not used during training for different sequence lengths and initial damages. Ultimately, the trained erosion model is the backbone of the framework but it can have several different application depending on the inputs of the user.

---

## Author Comment (AC2)

**Response to Anonymous Referee #2 comments of Manuscript ID WES-2022-55 entitled "Introducing a data-driven approach to predict site-specific leading edge erosion"**

Thank you for taking the time to review our article. We have addressed your comments attentively, for which the details are provided below.

1. A comparison between the minimal feature approach followed in this paper and more expensive data-driven approaches mentioned in related work is needed to understand how the "data minimization" affects the quality of predictions.

   An exhaustive data-driven feature selection approach was taken and it was found that by introducing more features could slightly increase the statistical performance but the physicality of the model was reduced, i.e., the physical criteria regarding interpretability (specified towards the end of Section 2.3) were violated. This was already stated in the manuscript but maybe a bit too vague. For that reason, the manuscript has been updated to clarify this better and now reads as:

   "For this reason, using more input features would simply require a larger dataset. It should be mentioned, that an exhaustive feature selection was performed using brute-force evaluation of feature subsets. In the end, it was found that by introducing more features, the explainability of the model was reduced even if the statistical performance was slightly improved. When working with such few training samples, choosing the optimal input features becomes a trade-off between accuracy and interpretability. Finally, it was chosen to use only the accumulated impingement and the initial damage as input features."

2. The choice of the ensemble model should be better justified and a comparison to other types of models from interpretable ones like decision trees and linear regression models to black-box models should be added. Interpretability was mentioned as a requirement when discussing dimensionality (around line 255) so an interpretable model might reveal further insights regarding the important features for different damage types

   Since we are working with only two input features, the response surface of the model can actually be visualized as shown in Figure 14. Here we can clearly interpret how the input features affect the output. When using such few input features, the interpretability of the model can to some degree be maintained even for a black-box model like the FFNN. However, the authors agree that the model selection could be better described. For that reason, the section regarding this has been updated and now reads as:

   "It was chosen to use a simple feedforward neural network (FFNN) as the weak learners for the ensemble model. Several models were evaluated as candidates by means of the validation described previously, i.e., statistical performance and physical interpretability. The neural network was found to be the best model when comparing error statistics. As an example, the root-mean-square-error was found to be 30.0 % lower than the support-vector-machine, 36.4 % lower than polynomial chaos expansion (PCE) of first order (46.2 % for second order) and 41.7 % lower than decision trees. For other error metrics the same behavior was observed. In addition, the neural network was also found best suited for capturing physical characteristics such as the incubation period. Specifically, it can be mentioned that PCE up to third order was evaluated in an exhaustive manner, i.e., by evaluating all the polynomial expansions repetitively. It was found that a first-order model performed best whereas higher-order models failed to represent the physicality requirements specified earlier.

   When using ensemble techniques such as bagging, it is desired to use a high-variance model which is able to learn non-linear relations. This is because the variance is reduced through bagging and the bias is increased (or maintained). This trade is not unique for neural networks and other models such as those based on decision trees or expansion models can have the same ability.

   Finally, it can also be mentioned that the ensemble training process used in this study, requires extrapolation in some of the data splits when validating against the test data, and though extrapolation is never recommended for ML applications, the FFNN has ability to do it anyway and performed better than SVM or models based on decision trees. It is the inherent definition of decision trees that makes them unsuited for extrapolation, e.g., the output of a decision tree is limited by the leaf nodes and can therefore never exceed the outer leaf nodes."

3. Several points need further clarifications:

   - Sparsity (Section 2.3) refers to missing data or missing labels? Please expand the discussion on "sparsity and limitations in data availability" and provide e.g. % of missing data.

   - it is not clear what the cardinalities of the training, validation and testing sets are

- the input features should become clear from the very beginning. This information comes too late now and becomes crystal clear only in section 4. A better idea would be to summarize the features in a table, including their value domain.

Sparsity is used as a general term and refers to both the limited amount of blade inspections (number of samples) and the limited operational SCADA data and mesoscale channels (number of inputs to the impingement model). Missing periods or invalid data was briefly mentioned but it has been slightly extended such that it reads as:

"To account for any missing periods or invalid data, the accumulated rain impingement was scaled with a factor corresponding to the ratio between the theoretical length of the time series and the actual available length of the time series. Generally, the scaling factor was found to be close to one, indicating a good availability with no missing periods or invalid data. However, since the mesoscale weather data was only available from May 2013, some operational intervals will not be fully covered. The is specifically the case for one of the wind farms which was commissioned in 2009. Therefore, the temporal coverage by the mesoscale weather data is only partial and relies on linear scaling."

For the target features (blade inspections) such figures are kept confidential for this article, but the training is performed considering the available periods only.

The splitting strategy is described in Section 2.3, now updated as:

"Here in this study with 18 samples in total, we evaluated 16/2 (i.e., 16 training, 2 test samples), 15/3, and 14/4 split configurations, corresponding to a testing portion of 11%, 17%, and 22%, respectively."

Figure 1 of the article very early on aims to provide a high-level description of input and target features for both training and application. The description of this figure is now updated to list these features more explicitly, and provide further details on the specific channels included in the workflow.

4. The organization of the paper needs improvement. The introduction section is too long and also covers related work, part of which is also discussed in section 4. I suggest a separate related work section. Section 2.3 is also too long and could be better organized into e.g., data, model and parameter tuning. The discussion section is very interesting but too long to follow. I suggest you split it into different subsections regarding e.g., feature/data choices, model choices, experimental findings etc. Also the discussion includes suggestions for future extensions, the title therefore should be changed accordingly.

It is a standard practice to include the relevant (and previous work) in the Introduction section - only a selected few is re-visited in the discussions on Section 4. Therefore, the authors would like to avoid defining an additional section regarding 'related work'. For the organization of the discussion section itself though, the authors agree to reviewers feedback and the section is further organized via several subsections/paragraphs, to have a better flow for the interested reader. The section title is also updated as 'Discussions and Future Work'. For Section 2.3, the authors added paragraph identifiers to help clarify the information provided therein.

5. I believe the novelty of this work is not the ML model but rather the minimal data approach that is followed to train such a model. The title of the paper should therefore be updated.

The authors agree that the novelty of the study is not the ML itself but rather the development of a framework which maps the mesoscale weather simulations to full-scale blade inspections at several sites, and allows for predicting site-specific erosion, aligned with the terminology used in the industry. Since the machine learning approach was not highlighted in the title to begin with, we find it still appropriate. (However, we have decided to underline the mapping of the weather simulations to inspections, hence the title is updated as "Introducing a data-driven approach to predict site-specific leading edge erosion from mesoscale weather simulations")

6. Figure 1 needs improvement, for example, the input features could be clearly indicated. Also, I find the current "ensemble splitting", "model selection", "training and validation", "ML trained model" components not very informative, I believe the input data should be clearly depicted. The training and testing parts are confusing, it seems that only the weather data are used during testing.

The purpose of Figure 1 is to give the reader a high-level overview of the whole framework to use as a guideline throughout the paper. Each step in the figure is described in details throughout the manuscript. For better understanding, the figure is also described more clearly in the beginning of section 2 and reads as:

"The overall workflow of the modeling framework can be found in Figure 1 showing the main steps. The workflow is two-folded and starts with the model training process. Here, the user is required to provide blade inspection, time series of mesoscale weather data (wind speed and rain rate) and wind turbine characteristics. The three data types should be coherent in the way that the mesoscale weather

data should cover the location and operational time interval of the inspected wind farm. Also, the wind turbine characteristics should match that of the inspected turbines. Time series of wind speed and rain rate are used together with the wind turbine characteristics to calculate the accumulated impingement over the operational time interval. This feature is used as the main input to the internal model. The observed defects from the blade inspections are encoded and ranked using a weighting scheme. Since the observed defects are accumulated throughout the operational time interval, the encoding allows for retrieving both an initial damage and an accumulated damage. The initial damage is used as an input to the internal model whereas the accumulated damage is used as the target.

The training is followed by the application of the trained model. Here, the user can input new weather data and specify wind turbine characteristics of new or existing wind farms. Similar, the user can specify the initial damage state to estimate erosion damages for new or existing wind farms. Since the output of the internal model is a site-specific damage prediction, the model is very flexible and can be utilized depending on the available data of the user, e.g., in the form of an interactive erosion map. A detailed description of each step in workflow will be given forthwith."

7. Statements like "simple neural networks are also well suited as weak learners for ensemble modeling, whereas simple linear regression models are not" (line 315) "they are able to interpolate and, to some extent, extrapolate, which is not the case for other machine learning classes such as support vector machines or those based on decision trees" (line 315) should be supported by appropriate references.

Bagging works well on high-variance models like NN or decision trees but on low-variance models like simple linear regression is does not work well. This is because the variance is reduced through bagging, but the bias is increased. Linear regression which for non-linear cases is high-biased, the bias would increase through bagging and reduce the accuracy. This statement is true in general but there exist cases where it does not apply. In our study, we did have some linear correlation between inputs and outputs, which is the reason why we did not get terrible results when using bagging for linear regression. However, the NN was still found to outperform the other models. A reference regarding the bias/variance trade-off has been added to the manuscript.

Regarding the ability to extrapolate, it is the inherent definition of decision trees that makes them unsuited for extrapolation, e.g., the output of a decision tree is limited by the leaf nodes and can therefore never exceed the outer leaf nodes. The ensemble training process used in this study, requires extrapolation in some of the data splits when validating against the test data, and though extrapolation is never recommended for ML applications, the FFNN has ability to do it anyway and performed better than SVM or models based on decision trees.

The additional description supplied for comment 2, should also cover this comment from the reviewer, including extra references to the updated version of the corresponding text.

---

## Author Response (AR2)

Dear Professor Athanasios Kolios,

Thanks a lot for your comments. Please find the reply for each below:

- *In the caption of Figure 5 please rephrase "are not non-structural defects".* Completed

- *I suggest to replace "Ensemble splitting" in page 15 with "Ensemble generation".* Completed

- *I suggest to replace "Validation" in page 16 as it is reserved term in ML pointing to train-validation-test split.* Completed – changed to "Performance verification".

- *In the validation section, please add a few more information on the synthetic data, how they were generated and by how many instances the training set was augmented.* Completed - an additional description of the generation of the synthetic data has been added and Figure 10 has been updated to clearly visualize the synthetic data.

- *I suggest to replace "Model selection" in page 17 with "Weak models".* Since it was suggested by one of the reviewers to add this subsection that refers more to algorithm selection, the subsection title has been changed to "Algorithm selection".

- *The title of section 3.1 could be more informative.* Now changed to "Ensemble model performance".

- *In the same section, it would be helpful for the end reader to organize the results based on the goal of the experiment. E.g., Figures 12 and 13 could be grouped together as both refer to the model performance w.r.t. ground truth. Figure 14 is about the "plausibility" of the predictions; I use the term "plausibility" here instead of "validity" (See my comment above on validation), though I am not sure if "plausibility" is the best term. Figure 15 is about the effect of sequence lengths and so on and so forth.* Figures 12 and 13 have been grouped together and suggested pointers have been added to the captions of now figures 12, 13 and 14.

- *You could use bold/italics etc to highlight some terms in text, this would be esp. useful for long sections like Section 4.* The take-home messages from section 4, as well as other sections, are summarized under conclusions anyway.

- *I would replace the "future work" in the title of section 4 with "open issues/challenges".* Now replaced by "Challenges".

Thank you very much.

Kind regards,

The Authors